# WHEN SHOULD AI ASK: DECISION-THEORETIC ADAPTIVE COMMUNICATION FOR LLM AGENTS

## ABSTRACT

Large language model (LLM) agents are increasingly used to assist people with complex tasks, but real-world user queries are often underspecified. When information is missing, agents face a dilemma: act autonomously and risk costly mistakes, or ask too many clarifying questions and frustrate the user. We propose a decision-theoretical framework for *adaptive communication* that dynamically determines when clarification is necessary based on three contextual factors: query ambiguity, task risk, and user cognitive load. Our approach instantiates this framework with a Value of Information (VoI) method that, at inference time, explicitly weighs the expected utility of clarification against its communication cost. Unlike existing confidence thresholds or heuristic prompting approaches, our method requires no task-specific tuning and adapts flexibly across domains and stakes. In experiments on 20 Questions, medical diagnosis, flight recommendation, and Webshop, our adaptive strategies consistently achieve higher utility than baselines, asking fewer unnecessary queries and requiring no hand-tuned thresholds. These results establish a principled foundation for building LLM agents that are not only competent actors, but also *strategic communicators* able to adapt their behavior to user context and task stakes for more reliable real-world collaboration.

## 1 INTRODUCTION

Most real-world user requests are underspecified: they carry unstated preferences, implicit context, and latent goals (Malaviya et al., 2025; Peng et al., 2024; Ma et al., 2024). Faced with this uncertainty, current LLM agents are caught in a dilemma: act prematurely and risk costly misalignment errors, or ask too many clarifying questions and frustrate the user. This balance between **autonomous action** and **active communication** is a key component to build reliable, collaborative AI systems.

This dilemma is not merely theoretical; it manifests in common user interactions, as illustrated in Figure 1. An agent tasked with "Buy me a flight to New York City" could act on incomplete information leading to a negative outcome (A). At the other extreme, the agent could interrogate the user about every possible preference—stops, airlines, departure times—causing fatigue and frustration (B). A truly intelligent agent must strike a middle path: inferring likely preferences from context while asking only a few strategic, high-impact questions to resolve the most critical uncertainties (C). This requires the agent to reason not just about what action to take, but also about when clarification is worth the user's effort.

We argue that enabling this strategic communication requires agents to reason about many context-dependent factors Fragiadakis et al. (2024). While a comprehensive model of collaboration could include numerous variables, We frame this as a decision-theoretic trade-off between acting and clarifying, which we model using three fundamental factors: (1) **Query Ambiguity**: the degree of uncertainty about the user's true intent; (2) **Task Risk**: the severity of the consequences of a wrong action; and (3) **Cognitive Load**: the cost, in time and effort, imposed on the user by asking for clarification. An agent booking a flight for a critical business meeting should be more cautious and communicative than one guessing an animal in a low-stakes 20 Questions game, even with the same level of uncertainty.

To operationalize this reasoning, we propose a decision-theoretic framework grounded in the Value of Information (VoI), a classic principle from decision theory Raiffa & Schlaifer (1961). Our inference-time method allows an LLM to explicitly calculate the expected utility gain of asking a potential

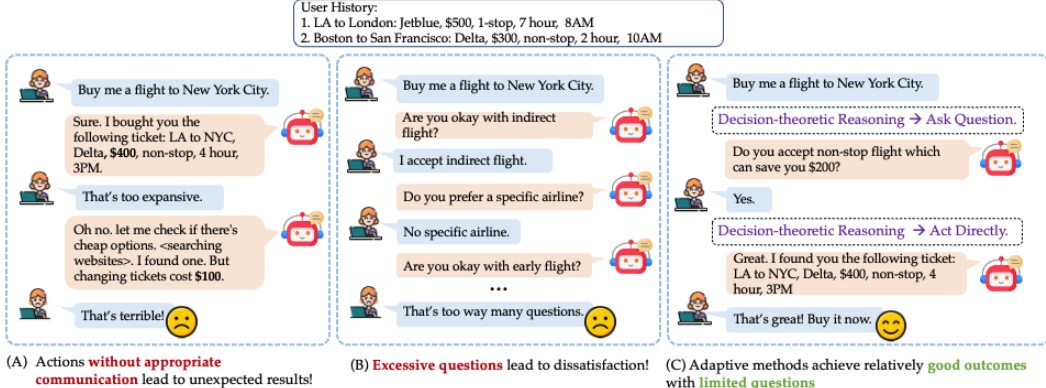

Figure 1: **Illustration of different communication methods and user reaction.** Given user flight history, an LLM agent is able to infer user latent preferences with some probability. Excessive questions that asks about every aspect of preference would lead to user dissatisfaction (A) while directly acting without communication could lead to unexpected consequences (B). Decision-theoretic reasoning can balance expected utility gain via asking user questions against communication cost to achieve efficient but effective communication at inference time (C).

question, weighing it directly against the communication cost. This provides a principled mechanism for the agent to decide whether the information it might receive is worth the user's attention. Our contributions are threefold: (a) We formalize the adaptive communication problem in human-agent interaction from a decision-theoretic perspective, identifying three key factors: ambiguity, risk, and cognitive load. (b) We propose a practical, inference-time VOI-based method that allows an LLM to estimate these contextual factors and dynamically decide whether to act or to seek clarifications (c) We demonstrate through experiments across four distinct domains: 20 Questions, medical diagnosis, flight booking, and online shopping, that our parameter-free VoI method automatically identifies the optimal operating point. In nearly all settings, it matches or exceeds the performance of the best-tuned baselines, which in contrast, require a brittle and impractical manual search over their hyperparameters for each specific task and cost structure.

## 2  RELATED WORK

**Standard LLM Agent Paradigm.**  Our work is situated within the broader context of developing autonomous LLM agents. Much foundational research in this area focuses on improving agent reasoning, planning, and tool-use capabilities. Prominent paradigms like ReAct (Yao et al., 2023) and others are often evaluated in benchmarks that, while complex, assume the user's initial instruction is complete and unambiguous (Yao et al., 2022; Zhou et al., 2023; Xie et al., 2024). This focus on execution rather than intent clarification leaves a critical gap. Recently, a new wave of research has begun to address agent reliability by introducing principled frameworks from decision theory (Liu et al., 2024; Lin et al., 2024; Chen et al., 2025). However, these approaches typically focus on making an optimal decision given a static, pre-defined state of information. Our work bridges these two areas: we adopt the rigor of decision theory but focus on the upstream problem of active information gathering, allowing the agent to dynamically resolve ambiguity before committing to an action.

**LLM Proactive Communication.**  Prior work has explored prompting techniques to improve LLM interactivity. These methods can elicit user preferences (Li et al., 2023) or encourage active disambiguation of ambiguous queries (Deng et al., 2023; Zhang et al., 2024b). While prompting can directly induce clarifying behaviors, prior work shows that the resulting strategies are often suboptimal without more principled planning or learning algorithms. Our work provides such a principled algorithm to govern the agent's communication decisions.

**Uncertainty-Gated and Information-Theoretic Methods.**  A more systematic approach uses model-uncertainty estimates to decide when to seek clarification, triggering a question when prediction confidence or entropy falls below a selected threshold (Wang et al., 2025; Zhang & Choi, 2023; Kuhn

et al., 2022; Ren et al., 2023). While an improvement over heuristics, these information-centric views can be insufficient, as they do not directly consider the downstream task's stakes. Our method addresses this by employing the Value of Information (VoI) (Raiffa & Schlaifer, 1961; Howard, 1966), a core concept from decision theory. Instead of measuring information gain in isolation, VoI measures how that information is expected to improve the utility of the final action, explicitly connecting the purpose of communication to the stakes of the decision.

**Learning-Based Approaches.** Different from the inference-time algorithms above, another line of research uses reinforcement learning to improve LLM collaboration with humans. Variants of Direct Preference Optimization (DPO) have been applied to encourage models to request clarification when needed (Zhang et al., 2024a; Chen et al., 2024; Wu et al., 2025; Andukuri et al., 2024). However, RL is often task-specific, requiring a carefully designed simulation environment and training pipeline, which is fundamentally different from our VOI-based method which operate purely at inference-time.

## 3 PROBLEM FORMULATION

We formulate the adaptive communication task as a sequential decision-making process where an LLM agent interacts with a user to select an optimal action.

**Preliminaries.** The agent receives an initial, potentially ambiguous, user query $S$. The user's true goals and preferences are represented by a latent state $\theta \in \Theta$, which is not directly observable by the agent. The agent has access to a set of possible terminal actions $a \in \mathcal{A}$. To resolve ambiguity about $\theta$ and choose the best action $a^*$, the agent can engage in a multi-turn dialogue with the user.

**The Clarify-or-Commit Process.** The interaction proceeds in a sequence of turns. At each turn $t$, given the dialogue history $H_t = (q_1, u_1, \ldots, q_{t-1}, u_{t-1})$, the agent must make a decision:

1. **CLARIFY:** Select and pose a question $q_t$ from a set of possible questions $\mathcal{Q}$. Upon receiving the user's answer $u_t$, the history is updated to $H_{t+1}$ and the process continues.

2. **COMMIT:** Terminate the dialogue and select a final action $a \in \mathcal{A}$ based on the current history $H_t$.

The agent's strategy for making this choice at each turn is the **clarify-or-commit** policy, which is the central object of our study. This simple clarify-or-commit choice lies at the heart of adaptive communication: every question carries both the potential to reduce uncertainty and the cost of additional user effort.

**Utility and Objective.** The success of a committed action $a$ is measured by a utility function $U(\theta, a)$, which quantifies how well the action aligns with the user's true latent state $\theta$. Communication incurs a cost $c(H)$, representing the user's cognitive load, which quantifies the time and effort user spent on the dialogue. If the agent commits to action $a$ after a final history $H$, the total utility is $U(\theta, a) - c(H)$. The agent's objective is to devise a policy that maximizes the expected total reward, optimally balancing the utility gain from asking questions against cumulative communication cost.

## 4 METHODS

To address the clarify-or-commit problem formulated in the previous section, an agent requires a principled policy for deciding when the potential benefit of asking a question outweighs the cost of interaction. Simple heuristic-based strategies, such as leveraging model's confidence, often fail because they do not explicitly reason about the downstream consequences or the stakes of the decision. To overcome this limitation, we propose an adaptive policy grounded in the Value of Information (VoI), a core concept from decision theory Raiffa & Schlaifer (1961). Our approach calculates the expected utility gain of asking a potential question and proceeds only if this gain outweighs the communication cost.

## 4.1 Value of Information Method (VoI)

The baselines above are either non-adaptive or rely on generic, task-agnostic heuristics like confidence. They fail to explicitly reason about the *value* of the information a question might provide in the context of heterogeneous task stakes and unequal feature importance. For example, in web shopping, clarifying the **functionality and size** of a product is critical, while colors and minor accessories are less important. A confidence-only controller might waste user time on low-value attributes or stop early while uncertainty remains on high-weight features.

To address this, we propose an adaptive policy based on the **Value of Information (VoI)**. At each turn, this method calculates the expected utility gain of asking a potential question and proceeds only if this gain outweighs the communication cost.

**Beliefs and Expected Utility.** Given the user query and available actions, we prompt the LLM to generate k latent factors $\theta$ that it believe affect the decision and the belief $b(\theta)$, the probability distribution over $\theta$, following Liu et al. (2024). Given a belief $b(\theta)$, the expected utility of committing to an action $a$ is:

$$\text{EU}(a \mid b) = \mathbb{E}_{\theta \sim b}[U(\theta, a)] = \sum_{\theta \in \Theta} b(\theta) U(\theta, a). \tag{1}$$

If the agent were to commit immediately, it would choose the action $a^* = \arg\max_{a \in \mathcal{A}} \text{EU}(a \mid b)$. The utility of this decision is the value of acting under the current belief $b$:

$$V(b) = \max_{a \in \mathcal{A}} \text{EU}(a \mid b). \tag{2}$$

**Calculating the Value of a Question.** Before asking a question $q$, the LLM can simulate the potential outcomes. Let $y \in \mathcal{Y}$ be a possible user answer to $q$. The LLM can use its world knowledge to estimate the probability $p(y \mid q, b)$ of receiving answer $y$. For each possible answer $y$, the LLM would update its belief to a new posterior belief $b_y$ using Bayes' rule. The expected value of the decision *after* receiving an answer to question $q$ is then an expectation over all possible answers:

$$V_{\text{post}}(b, q) = \sum_{y \in \mathcal{Y}} p(y \mid q, b) \cdot V(b_y). \tag{3}$$

The **Value of Information** for question $q$ is the difference between the expected utility after asking and the utility of acting now:

$$\text{VoI}(q) = V_{\text{post}}(b, q) - V(b). \tag{4}$$

**The Clarify-or-Commit Policy.** Our framework uses this VoI calculation to establish a decision rule. At each turn, the agent evaluates the net utility gain for each candidate question:

$$\text{NetVoI}(q) = \text{VoI}(q) - c, \tag{5}$$

where $c$ is the per-question communication cost. The agent then selects the question $q^*$ with the highest positive net value. If no question has a positive net value ($\max_q \text{NetVoI}(q) \leq 0$), it means the expected utility gain from further communication is not worth the cost. At this point, the agent terminates the dialogue and commits to the best action under its current belief. A detailed, practical algorithm is presented in Algorithm 1. The prompt we used can be found in Appendix A.3.

## 5 Experimental Setup

### 5.1 Baseline Methods

**No-Question.** This baseline represents the standard agent paradigm. Given the initial query $S$, the agent commits to an action immediately without any communication with the user. It relies solely on its initial understanding of the user's intent.

**Fixed-Round.** This non-adaptive baseline asks a fixed number of $k$ questions before committing to an action. It serves to isolate the benefit of interaction from the benefit of *adaptive* interaction by exploring a fixed trade-off between information gathering and communication cost.

---

**Algorithm 1** VOI Algorithm

---

**Require:** Instruction $S$; action set $\mathcal{A}$; utility $U(\theta, a)$; question generator $\mathrm{GenQ}$; belief updater
    $\mathrm{Update}$; cost $c(\cdot)$; clarification budget $K_{\max}$
1: $H \leftarrow \{S\}; \quad b \leftarrow \mathrm{Prior}(S)$
2: **for** $t = 1, 2, \ldots, K_{\max}$ **do**
3:     $Q \leftarrow \mathrm{GenQ}(H)$                                    ▷ small set of targeted questions
4:     $V_0 \leftarrow V(b) = \max_{a \in \mathcal{A}} \mathbb{E}_{\theta \sim b}[U(\theta, a)]$
5:     **for all** $q \in Q$ **do**
6:         Sample plausible replies $\{(y_k, \pi_k)\}_{k=1}^{K}$ from $P(\cdot \mid b, q)$
7:         $V_q \leftarrow \sum_{k=1}^{K} \pi_k V\big(\mathrm{Update}(b, q, y_k)\big)$
8:         $\mathrm{VoI}(q) \leftarrow V_q - V_0 - c(q)$
9:     **end for**
10:    $q^* \leftarrow \arg\max_{q \in Q} \mathrm{VoI}(q)$
11:    **if** $\mathrm{VoI}(q^*) \leq 0$ **then break**                       ▷ clarification not worthwhile
12:    **else**
13:        Ask $q^*$, observe $y$; $H \leftarrow H \cup \{(q^*, y)\}$; $b \leftarrow \mathrm{Update}(b, q^*, y)$
14:    **end if**
15: **end for**
16: **return** $a^* \in \arg\max_{a \in \mathcal{A}} \mathbb{E}_{\theta \sim b}[U(\theta, a)]$                ▷ final commitment

---

**Adaptive Prompting.** This baseline prompts the LLM to reason about whether it feels confident enough to act or if it should ask a question. The number of questions is not predetermined, but the decision to stop is based on the model's heuristic self-assessment rather than a formal criterion.

**Confidence Thresholding.** This adaptive baseline formalizes the heuristic of Adaptive Prompting. The agent continues to ask questions as long as its predictive confidence in the best action $a^*$ remains below a tunable threshold $\tau$. We measure confidence using the model's verbalized confidence scores (Tian et al., 2023), a common practice for modern LLMs. This method is adaptive, but crucially, the threshold $\tau$ must be manually tuned for each task and cost setting to achieve optimal performance.

## 5.2 TASKS AND MODELS

**Mixed-Stakes 20 Questions.** The 20 Questions game is a classic guessing game with a long history as a paradigm for studying human and artificial decision-making under uncertainty. It provides a controlled environment to test how an agent performs strategic information gathering. Following the setup of Hu et al. (2024), the agent must identify a target concept from a known candidate set by asking a series of binary (yes/no) questions. Our key modification is to explicitly test how the agent adapts to varying **task risk**. We create two parallel versions of this task:

- **Low-Stakes (Animal Guessing):** The agent identifies an animal from a set of 100. A correct guess yields a terminal utility of $U = 1$.

- **High-Stakes (Medical Diagnosis):** The agent diagnoses a medical condition from a set of 15 diseases, using real doctor-patient chat histories as input. A correct diagnosis yields a utility of $U = 10$.

**Flight Recommendation** We adopt a task designed to model the elicitation of multi-faceted user preferences, a common challenge in real-world assistants. Our setup is inspired by the recent work of Qiu et al. (2025) is derived from the FLIGHTPREF dataset originally proposed by Lin et al. (2022). The agent is presented with a user's choice history over five rounds of flight selections. In a final, held-out round, the agent must predict which of three new flight options the user will prefer. Each flight is defined by 8 features (e.g., price, stops, airline), and each user has a latent reward function defining their preferences over these features. The agent can ask clarifying questions to uncover these preferences before making its final prediction. This task tests the agent's ability to strategically query a complex, multi-attribute preference space to infer a user's reward model from their contextual choices. The agent's prediction for the new round will be scored based on this reward function.

**Ambiguous WebShop**   To test our agent in a more realistic, interactive environment, we adapt the WebShop benchmark (Yao et al., 2022). In the original setting, user instructions are created to be relatively well-specified (e.g., "buy a red Adidas t-shirt, size medium"). We deliberately introduce **query ambiguity** by removing details from the user's request (e.g., "buy a t-shirt") to simulate underspecified real-world user query. The agent must then decide whether to act on this partial information (e.g., `search("t-shirt")`) or to ask clarifying questions about attributes like size, color, or brand. This task evaluates the agent's ability to balance autonomous web navigation with strategic information gathering to resolve under-specified user requests.

**Models**   We consider a selection of leading LLMs to evaluate the performance of our proposed method, including GPT-4.1 (OpenAI, 2025) and Gemini-2.5-Flash (Comanici et al., 2025).

## 6   RESULTS

### 6.1   MAIN RESULTS

Our central findings are summarized in Figure 2. Across all tasks and communication cost settings, our VoI-based agent consistently achieves state-of-the-art utility. Crucially, it does so without requiring task-specific threshold tuning, showcasing its robustness and practical advantages.

**VoI excels by finding the optimal utility-cost balance.**   As shown in Figure 2, our VoI agent (starred marker) consistently ranks as the top-performing method across the Mixed 20Q, Flight Recommendation, and Ambiguous WebShop tasks. For instance, in Mixed 20Q with a communication cost of $c = 0.01$, VOI achieves a utility of 14.14, significantly outperforming the best-tuned confidence-thresholding baseline (11.49 at $\tau = 0.90$). This performance advantage stems from VOI's ability to dynamically determine the optimal number of clarification questions, a stark contrast to fixed-round and confidence-based methods that require brittle, manual tuning of a threshold for each specific task and cost structure.

**Adaptive communication is essential for ambiguous tasks.**   The "No Question" baseline establishes the necessity of proactive communication. On the Mixed 20Q task, where the initial query is inherently underspecified, this baseline's accuracy is near zero for both low-stakes (animal) and high-stakes (medical) variants. However, as shown in Figures 2(f) and 2(l), when communication costs are prohibitively high, avoiding questions becomes a competitive strategy. In these scenarios, our VOI method correctly adapts by stopping communication early, demonstrating its ability to gracefully handle the full spectrum of cost-benefit scenarios.

**Adaptive prompting are insufficient for robust performance.**   The Adaptive Prompting baseline shows that simply instructing an LLM to "ask questions when needed" offers an improvement over non-adaptive strategies. However, its performance is inconsistent and consistently lower than more structured methods. This is because the decision to communicate is based on the model's uncalibrated, internal "feeling" of confidence, rather than a formal criterion. It lacks a principled mechanism to weigh the potential information gain against the explicit communication cost, leading to suboptimal and unpredictable behavior.

**Fixed-round communication strategies are fundamentally suboptimal.**   A fixed-round policy, which asks a predetermined number of questions, fails to adapt to the specific needs of a given query. As illustrated in the inverted-$U$ shape of the "Fixed Round" curves in Figure 2, utility initially increases with more questions but then declines as communication costs overwhelm the benefits of additional information. The optimal number of questions varies significantly with the task and cost, highlighting the necessity of an adaptive policy.

**Confidence thresholding is effective but brittle.**   The confidence thresholding baseline provides a strong, adaptive competitor. With the *correctly* tuned confidence threshold $\tau$, its performance can be comparable to our VOI method (e.g., on GPT-4 for Mixed 20Q and Webshop). However, this effectiveness is its Achilles' heel; the optimal $\tau$ is highly sensitive and must be manually selected for each task and cost combination, making it impractical for real-world deployment. Our VoI method

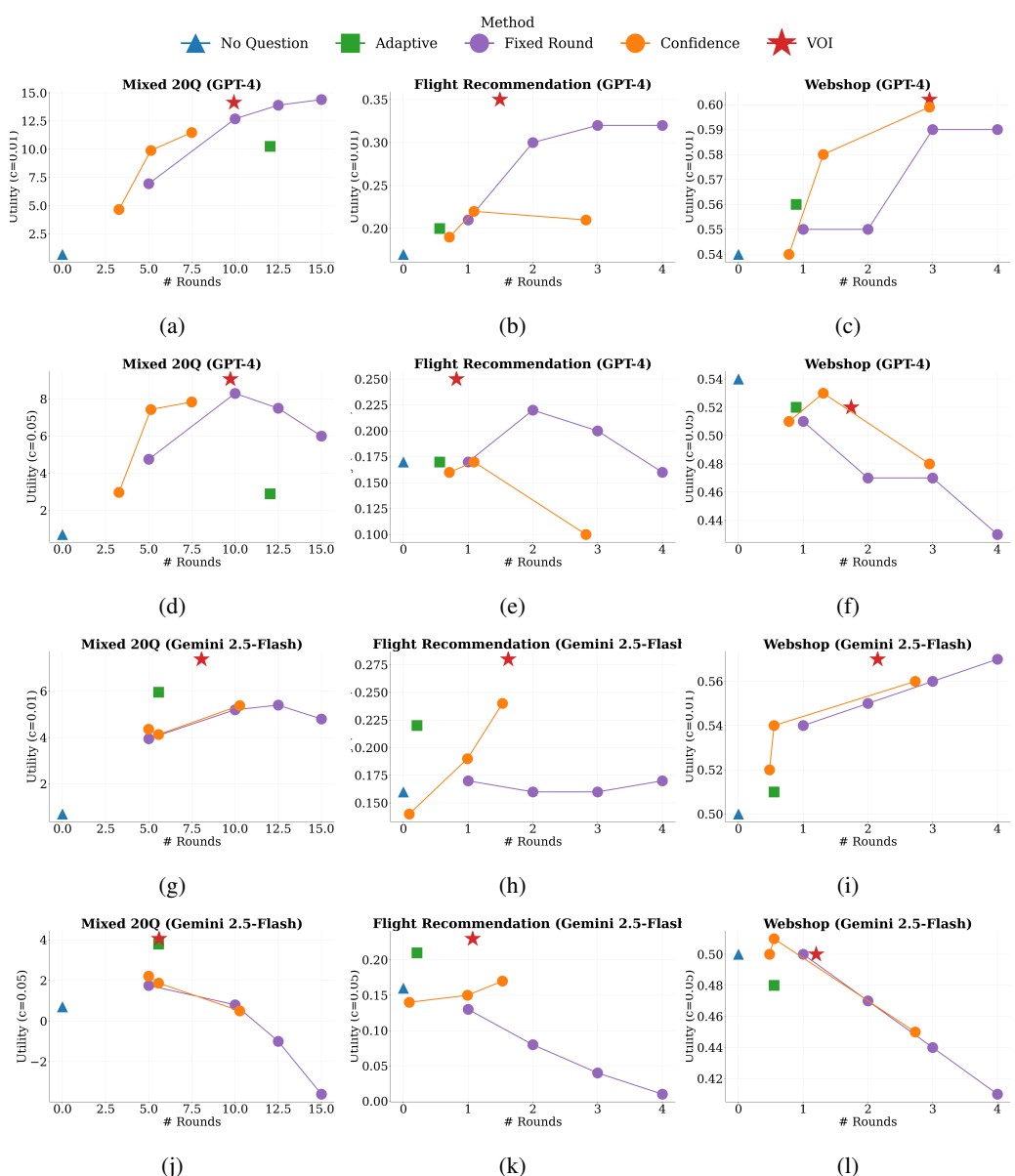

Figure 2: **Utility vs. Communication Rounds.** Final utility as a function of the number of clarification questions asked across our three tasks, for GPT-4 (top two rows) and Gemini-2.5-Flash (bottom two rows), with communication costs $c = 0.01$ and $c = 0.05$. Utility is defined as $U(\theta, a) - T \cdot c$. The curves for Fixed Round and Confidence Thresholding represent Pareto frontiers generated by varying their respective hyperparameters ($k$ and $\tau$). In contrast, our VoI agent (starred) is a parameter-free method. In nearly all settings, VoI automatically identifies an operating point that matches or exceeds the performance of the best-tuned baseline, demonstrating its superior adaptability and practical value.

provides a principled solution that matches or exceeds this performance without any such manual tuning.

## 6.2 ABLATION STUDY

**Ablation on Communication Cost.** As shown in Table 1, across the cost sweep on Mixed 20-Question the VoI controller matches or exceeds the strongest grid-searched baselines. We tune four baselines over nine threshold settings, and while the best baseline shifts with the communication

Table 1: **VOI vs. Baselines Across Costs (Gemini-2.5-Flash, Mixed 20 Question)**. This table compares the VOI policy's expected reward ($r_{\text{VOI}}$) against the best and second-best baselines via grid searching over 9 values. The $\Delta$ columns report VOI's margin over each baseline (positive means VOI is better).

| Cost | Best Baseline | $r_{\max}$ | Second Best | $r_{\text{second}}$ | $r_{\text{VOI}}$ | $r_{\text{VOI}}-r_{\max}$ | $r_{\text{VOI}}-r_{\text{second}}$ |
|------|---------------|------------|-------------|---------------------|------------------|---------------------------|-------------------------------------|
| 0.01 | Confidence ($\tau$=0.9) | 8.30 | Round ($\tau$=15) | 8.10 | 8.64 | 0.34 | 0.54 |
| 0.02 | Confidence ($\tau$=0.9) | 6.88 | Confidence ($\tau$=0.9) | 6.80 | 7.72 | 0.84 | 0.92 |
| 0.05 | Round ($\tau$=5) | 3.65 | Confidence ($\tau$=0.5) | 3.64 | 5.01 | 1.36 | 1.37 |
| 0.10 | Confidence ($\tau$=0.5) | 2.28 | Round ($\tau$=5) | 0.90 | 1.38 | -0.90 | 0.48 |
| 0.20 | No Question | 0 | Round ($\tau$=5) | -4.60 | -0.96 | -0.96 | 3.64 |

cost, VoI consistently selects an appropriate number of questions thst match the performance of the best baseline. Importantly, this pattern is stable across different choice of communication costs: VoI adapts smoothly to the stated cost rather than hinging on a brittle threshold choice.

**Calibration Analysis.** A critical component of our VoI framework is the LLM's ability to estimate a belief distribution $b(\theta)$ over latent user states. To analyze it, ideally we should compare model predicted distribution to the ground truth distribution. However, in the absence of the ground truth distribution for our tasks, we instead measure the argmax from the distribution against the ground truth item as the standard calibration analysis to approximate its distribution estimation accuracy. As shown in Figure 3, The results reveal that models are reasonably calibrated in Animal Guessing game but less calibrated for Medical Diagnosis which we suspect because of the inherent complication and noise in the symptoms of diseases. Despite this, we see that VOI are empirically effective and robust that consistently matches if not perform the best baselines after searching hyperparameters. We believe that current and future work that could improving model calibration under missing context Li et al. (2025) could further improve the performance of VOI.

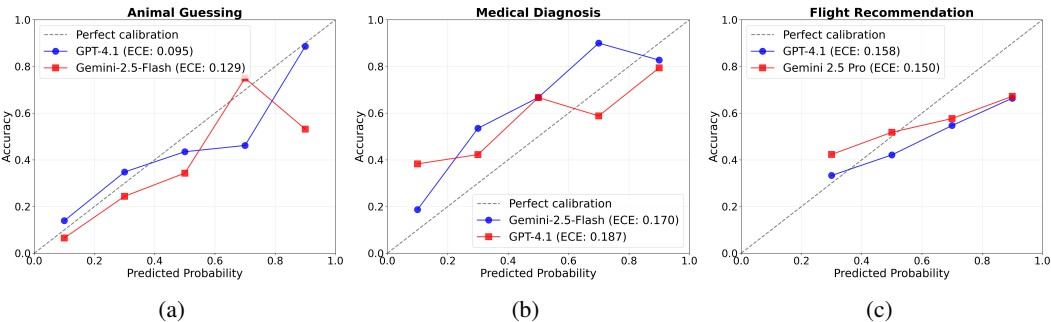

(a)                                          (b)                                          (c)

Figure 3: **Calibration Analysis** The figure presents the calibration analysis of GPT-4 and Gemini-2.5-Flash on Animal Guessing, Medical Diagnoiss, and Flight Recommendation. (In (c) the accuracy for predicted probability between 0 and 0.2 is omitted because very few samples fall in that range.

## 6.3 CASE STUDY: VOI IS RISK-AWARE

Figure 4 provides a compelling qualitative example of why the VoI framework is superior to heuristic-based methods like confidence thresholding. The experiment contrasts a low-stakes task (guessing an animal, reward=1) with a high-stakes task (medical diagnosis, reward=10), using an identical communication cost ($c = 0.05$).

In the **high-stakes medical diagnosis** (Fig. 4b), the potential reward for a correct answer is high. The VoI agent correctly calculates that even questions with moderate information gain are valuable enough to outweigh the communication cost. It, therefore, continues to ask clarifying questions until it is highly confident, stopping several rounds *after* the confidence-thresholding baseline would have stopped, even though significant ambiguity remains, leading to an incorrect diagnosis.

In the **low-stakes animal guessing game** (Fig. 4a), the maximum potential utility is low. Here, the VoI agent correctly assesses that the potential utility gain from asking many questions is not

| Dialogue | Conf. | VOI | Prediction |
|---|---|---|---|
| Q1: Is the animal a mammal?
A1: Yes | 5% | 0.26 | Elephant ✗ |
| Q2: Is the animal primarily found on land?
A2: Yes | 21% | 0.22 | Elephant ✗ |
| Q3: Is the animal larger than human?
A3: No | 41% | 0.20 | Otter ✗ |
| … | … | … | … |
| Q8: Is the animal native to Australia?
A8: No | 55% | 0.05 ⚠ | Alpaca ✅ |
| … | … | … | … |
| Q17: Is the animal known for its long neck?
A17: Yes | 90% ⚠ | -- | Alpaca ✅ |

| Dialogue | Conf. | VOI | Prediction |
|---|---|---|---|
| Q1: Are you experiencing any abdominal pain?
A1: Yes | 10% | 0.6 | Appendicitis ✗ |
| Q2: Do you have any nausea or vomiting?
A2: No | 30% | 0.34 | Irritable Bowel Syndrome ✗ |
| Q3: Have you noticed changes in bowel movements?
A3: Yes | 60% | 2.15 | Irritable Bowel Syndrome ✗ |
| … | … | … | … |
| Q9: Is pain in the lower left side?
A9: Yes | 90% ⚠ | 0.70 | Irritable Bowel Syndrome ✗ |
| … | … | … | … |
| Q13: Recent weight loss or loss of appetite?
A13: No | -- | 0.04 ⚠ | Constipation ✅ |

Figure 4: A side by side comparison for different methods for Mixed 20 Question task. The figure contrasts four controllers—No-Ask, Fixed-Round, Confidence Thresholding ($\tau = 0.90$), and our VOI policy—on a single Mixed 20Q instance with communication cost $c = 0.05$. Task stakes are encoded directly in the terminal utility: a correct animal guess yields reward 1 (low stakes), whereas a correct medical diagnosis yields reward 10 (high stakes). The objective maximizes decision utility minus dialogue cost, $U(\theta, a) - c(\xi)$.

worth the cumulative communication cost. It, therefore, halts the conversation earlier than the confidence-thresholding method, avoiding unnecessary cognitive load on the user for a low-risk task. The confidence-based agent, blind to the low stakes, would have continued asking questions, needlessly imposing cognitive load on the user for a trivial task.

This case study reveals that effective communication requires balancing two distinct pressures: the drive to reduce uncertainty (an epistemic goal) and the need to consider the task's stakes (a utilitarian goal). Confidence-based methods address only the former. The VoI framework excels because it naturally unifies both: it quantifies the value of reducing uncertainty precisely in terms of its expected impact on the final, stake-weighted utility. This principled balance enables the agent to be appropriately cautious in high-stakes scenarios and efficient in low-stakes ones—a critical capability for building trustworthy and effective human-AI collaborators.

## 7 CONCLUSION

Current LLM agents are often designed for well-specified tasks, leaving them brittle when faced with the inherent ambiguity of real-world user requests. In this work, we argued that overcoming this limitation requires agents to move beyond simple execution and develop a principled strategy for adaptive communication. We proposed a formal framework for this problem, centered on balancing three key factors: query ambiguity, task risk, and user cognitive load. Our primary contribution is a practical, inference-time method based on the Value of Information (VoI) that operationalizes this framework. By explicitly calculating the expected utility gain of a potential question and weighing it against its communication cost, our VoI-driven agent decides when to act and when to ask. Extensive experiments across diverse domains—including medical diagnosis and online shopping—demonstrate that our approach consistently outperforms non-adaptive and heuristic-based baselines. Crucially, it achieves this without the need for the brittle, task-specific threshold tuning that plagues other adaptive methods. Ultimately, this work provides a principled foundation for building LLM agents that are not just capable executors, but also thoughtful communicators. By equipping agents with a formal understanding of when information is valuable, we can create more aligned, efficient, and truly collaborative human-AI systems.

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

# A APPENDIX

## A.1 LIMITATION AND FUTURE WORK

Our framework provides a robust foundation for adaptive communication in LLM agents. The following represent deliberate scoping choices designed to isolate our core contribution and highlight clear avenues for future research.

**Scope of Interaction: Decision vs. Generation.** Our work deliberately focuses on the core decision of *when* to communicate, rather than *what* questions to generate. To this end, our experiments utilize a predefined set of actions ($a \in \mathcal{A}$) and clarifying questions, a methodological choice consistent with prior work (Hu et al., 2024; Kobalczyk et al., 2025). This controlled setting is critical for isolating the performance of our VoI-based *selection policy*, providing an unambiguous evaluation of our central claim. By controlling for the quality of question generation, we demonstrate the effectiveness of the decision-making principle itself. While extending this framework to fully open-ended dialogue is an important next step, establishing this robust selection principle is the necessary and foundational prerequisite. Our work thus provides the core engine around which more sophisticated generative components can be built.

**Model of Communication Cost.** We employ a linear communication cost model ($c(H) = T \cdot c$). Accurately modeling the nuances of human cognitive load is a major, open research challenge in its own right, spanning HCI and cognitive science. Therefore, in line with common practice in decision-theoretic analyses, we adopt a simplified and interpretable cost function. This allows us to clearly illustrate the fundamental trade-off between utility gain and cost, without introducing confounding variables from a more complex, speculative cognitive model. Importantly, the VoI framework itself is agnostic to the form of the cost function; the core decision rule, $\text{VoI}(q) - c(H)$, can readily incorporate more complex models as they are developed. Our contribution is the robust decision engine, and the simplicity of our experimental cost function serves to highlight its power in the clearest possible terms.

## A.2 MAIN RESULTS IN TABLES

Figure 5: GPT-4: results for different methods and thresholds across three tasks. For **Webshop**, LLM is normalized by 10 and utilities are Util = LLM − #T × {0.01, 0.05}. Mixed 20Q utilities are recomputed per spec. Within each *method*, the best utility is underlined. The global best per task/cost is ***bold+italic*** and the second best is **bold**.

| Method | Mixed 20Q | | | | | | | Flight Rec. | | | | | Webshop | | | | |
|---|---|---|---|---|---|---|---|---|---|---|---|---|---|---|---|---|---|
| | $\tau$ | Acc. (Animal) | Acc. (Med) | #T (Animal) | #T (Med) | Util. (0.01) | Util. (0.05) | $\tau$ | Reward | #T | Util. (0.01) | Util. (0.05) | $\tau$ | LLM | #T | Util. (0.01) | Util. (0.05) |
| No Question | – | 0.01 | 0.06 | 0.00 | 0.00 | 0.70 | 0.70 | – | 0.17 | 0.00 | 0.17 | 0.17 | – | 0.54 | 0.00 | 0.54 | ***0.54*** |
| Adaptive | – | 0.68 | 0.53 | 17.80 | 6.254 | 10.26 | 2.89 | – | 0.20 | 0.56 | 0.20 | 0.17 | – | 0.57 | 0.89 | 0.56 | 0.52 |
| Fixed Round | 5 | 0.24 | 0.51 | 5.00 | 5.00 | 6.95 | 4.75 | 1.00 | 0.22 | 1.00 | 0.21 | 0.17 | 1.00 | 0.56 | 1.00 | 0.55 | 0.51 |
| | 10 | 0.60 | 0.78 | 10.00 | 10.00 | 12.70 | 8.30 | 2.00 | 0.32 | 2.00 | **0.30** | 0.22 | 2.00 | 0.57 | 2.00 | 0.55 | 0.47 |
| | 15 | 0.77 | 0.78 | 15.00 | 10.00 | 13.90 | 7.50 | 3.00 | 0.35 | 3.00 | ***0.32*** | 0.20 | 3.00 | 0.62 | 3.00 | 0.59 | 0.47 |
| | 20 | 0.87 | 0.78 | 20.00 | 10.00 | ***14.40*** | 6.00 | 4.00 | 0.36 | 4.00 | ***0.32*** | 0.16 | 4.00 | 0.63 | 4.00 | 0.59 | 0.43 |
| Confidence | 0.50 | 0.20 | 0.31 | 4.01 | 2.54 | 4.67 | 2.97 | 0.50 | 0.19 | 0.71 | 0.19 | 0.16 | 0.50 | 0.55 | 0.78 | 0.54 | 0.51 |
| | 0.70 | 0.45 | 0.60 | 5.68 | 4.56 | 9.89 | 7.43 | 0.70 | 0.23 | 1.09 | 0.22 | 0.17 | 0.70 | 0.60 | 1.31 | 0.58 | 0.53 |
| | 0.90 | 0.59 | 0.65 | 8.48 | 6.49 | 11.49 | 7.84 | 0.90 | 0.24 | 2.82 | 0.21 | 0.10 | 0.90 | 0.63 | 2.95 | ***0.60*** | 0.48 |
| VOI | 0.01 | 0.76 | 0.78 | 11.80 | 8.07 | 14.14 | ***9.10*** | 0.01 | 0.36 | 1.49 | **0.35** | ***0.28*** | 0.01 | 0.63 | 2.95 | ***0.60*** | 0.49 |
| | 0.05 | 0.74 | 0.78 | 11.46 | 7.99 | 13.97 | **9.07** | 0.05 | 0.29 | 0.82 | 0.29 | **0.25** | 0.05 | 0.61 | 1.74 | 0.59 | 0.52 |

Table 2: Gemini-2.5-Flash: results for different methods and thresholds across three tasks. Format is the same as Figure 5

| Method | | Mixed 20Q | | | | | | | Flight Rec. | | | | | Webshop | | | |
|---|---|---|---|---|---|---|---|---|---|---|---|---|---|---|---|---|---|
| | $\tau$ | Acc. (Animal) | Acc. (Med) | #T (Animal) | #T (Med) | Util. (0.01) | Util. (0.05) | $\tau$ | Reward | #T | Util. (0.01) | Util. (0.05) | $\tau$ | LLM | #T | Util. (0.01) | Util. (0.05) |
| No Question | – | 0.01 | 0.06 | 0.00 | 0.00 | 0.70 | **0.70** | – | 0.16 | 0.00 | 0.16 | 0.16 | – | 0.50 | 0.00 | 0.50 | **0.50** |
| Adaptive | – | 0.28 | 0.37 | 4.78 | 6.36 | 5.96 | **3.79** | – | 0.22 | 0.21 | 0.22 | 0.21 | – | 0.51 | 0.55 | 0.51 | 0.48 |
| Fixed Round | 5 | 0.16 | 0.29 | 5.00 | 5.00 | 3.95 | 1.75 | 1.00 | 0.18 | 1.00 | 0.17 | 0.13 | 1.00 | 0.55 | 1.00 | 0.54 | **0.50** |
| | 10 | 0.33 | 0.30 | 10.00 | 10.00 | 5.20 | 0.80 | 2.00 | 0.18 | 2.00 | 0.16 | 0.08 | 2.00 | 0.57 | 2.00 | 0.55 | 0.47 |
| | 15 | 0.40 | 0.30 | 15.00 | 10.00 | 5.40 | -1.00 | 3.00 | 0.19 | 3.00 | 0.16 | 0.04 | 3.00 | 0.59 | 3.00 | **0.56** | 0.44 |
| | 20 | 0.39 | 0.30 | 20.00 | 10.00 | 4.80 | -3.60 | 4.00 | 0.21 | 4.00 | 0.17 | 0.01 | 4.00 | 0.61 | 4.00 | **0.57** | 0.41 |
| Confidence | 0.50 | 0.22 | 0.27 | 4.87 | 5.12 | 4.36 | 2.21 | 0.50 | 0.14 | 0.09 | 0.14 | 0.14 | 0.50 | 0.52 | 0.48 | 0.52 | 0.50 |
| | 0.70 | 0.16 | 0.31 | 5.06 | 6.08 | 4.13 | 1.87 | 0.70 | 0.20 | 0.99 | 0.19 | 0.15 | 0.70 | 0.54 | 0.55 | 0.54 | **0.51** |
| | 0.90 | 0.36 | 0.30 | 11.28 | 9.25 | 5.38 | 0.50 | 0.90 | 0.25 | 1.53 | 0.24 | 0.17 | 0.90 | 0.59 | 2.73 | **0.56** | 0.45 |
| VOI | 0.01 | 0.28 | 0.55 | 8.48 | 7.63 | 7.38 | 3.68 | 0.01 | 0.30 | 1.62 | 0.28 | 0.22 | 0.01 | 0.59 | 2.15 | 0.57 | 0.48 |
| | 0.05 | 0.15 | 0.50 | 4.20 | 6.99 | **6.01** | **4.05** | 0.05 | 0.28 | 1.07 | **0.27** | **0.23** | 0.05 | 0.56 | 1.20 | 0.55 | **0.50** |

## A.3 PROMPTS

## A.4 MIXED 20 QUESTIONS

---

**Animal — Direct Prompting**

You are playing 20 Questions as the guesser. Your goal is to figure out what animal I'm thinking of by asking questions.
You have asked {question_count} questions so far and have {remaining_questions} questions left.
The possible animals you're trying to guess include: {answer_set}
**Rules:**
1. Ask only yes/no questions (answerable with "Yes"/"No").
2. Ask one question at a time.
3. Keep asking until you use all 20 questions.
4. Do not ask the same question twice.
5. Do not guess a specific animal early (e.g., "Is it a cat?"). Start broad to narrow options.

---

Figure 6: Direct Prompting (Animal 20 Question)

---

**Animal — Auto Stop**

You are playing 20 Questions as the guesser. Your goal is to figure out what animal I'm thinking of.
You have asked {question_count} questions so far and have {remaining_questions} questions left.
The possible animals you're trying to guess include: {answer_set}
**Rules:**
1. Ask only yes/no questions.
2. Ask one question at a time.
3. When you're ready to guess, output: `My guess is: [animal].`
4. Do not ask the same question twice.
5. Avoid premature specific guesses; use broader categories to narrow down.

---

Figure 7: Auto Stop (Animal 20 Question)

## A.5 FLIGHT RECOMMENDATION

**Animal — Confidence Thresholding**

You are tasked with guessing the animal based on the entire conversation. Provide a final guess, even if uncertain. The set of possible animals is: {answer_set}.
Also estimate your confidence from 1 to 100 (100 = highest).
**Output exactly:** `{"guess": "<animal>", "confidence": <number 1--100>}`
**Confidence Scale (1–100):**
1–20: Very minimal evidence.
21–40: Weak evidence.
41–60: Moderate evidence.
61–80: Strong evidence.
81–100: Very strong evidence.

Figure 8: Confidence Thresholding (Animal 20 Question)

**Animal — VOI: Question Generation**

I'm playing a game of 20 Questions to identify an animal. Based on previous questions and answers:
{previous_qa}
Generate 5 yes/no questions that help identify the animal.
Format as a numbered list.

Figure 9: VOI: Question Generation (Animal 20 Question)

**Animal — VOI: Batch Answer Simulation**

You are playing a game of Twenty Questions. You will receive a yes/no question and a list of animals.
Answer for each animal with: **Yes** (clearly correct), **No** (clearly incorrect), or **Maybe** (uncertain/ambiguous/varies by species or not confident). Prefer "Maybe" over guessing.
**Format exactly as:** `Animal Name:  Yes/No/Maybe`
**Question:** {question}
**Animals:** {candidate_list}

Figure 10: VOI: Batch Answer Simulation (Animal 20 Question)

**Medical — Direct Prompting**

You are the doctor and must diagnose the patient using only yes/no questions.
You have asked {question_count} questions so far and have {remaining_questions} left.
Possible diagnoses: {answer_set}
You may ask up to 20 yes/no questions to understand the condition. At the end, output your diagnosis.

Figure 11: Direct Prompting (Medical Diagnosis)

---

**Medical — Auto Stop**

You are the doctor and may ask up to 20 yes/no questions to diagnose the patient.
You have asked {question_count} questions so far and have {remaining_questions} left.
Possible diagnoses: {answer_set}
You can ask up to 10 yes/no questions. Stop when you have enough information.
**Format your guess as:** `My guess is:  [diagnosis].`

---

Figure 12: Auto Stop (Medical Diagnosis)

---

**Medical — Confidence Thresholding**

Diagnose the patient based on the entire conversation. Provide a final diagnosis, even if uncertain. Set of diseases: {answer_set}.
Also estimate your confidence (1–100).
**Output exactly:** `{"guess":  "<diagnosis>", "confidence":  <number 1--100>}`
**Confidence Scale (1–100):**
1–20: Extremely uncertain.
21–40: Weak evidence.
41–60: Moderate evidence.
61–80: Strong evidence.
81–100: Very strong evidence.

---

Figure 13: Confidence Thresholding (Medical Diagnosis)

---

**Medical — VOI: Question Generation**

I'm a doctor trying to diagnose a patient's condition through a series of questions. Based on symptoms and previous answers:
{previous_qa}
Generate 5 yes/no questions that most effectively narrow the possible conditions (roughly halving the set each time).
Focus on distinguishing symptoms, risk factors, or medical history.
Format as a numbered list.

---

Figure 14: VOI: Question Generation (Medical Diagnosis)

---

**Medical — VOI: Batch Answer Simulation**

I'm a medical diagnostician. Below is a yes/no question and a list of medical conditions.
**Question:** "{question}"
For each condition, answer with just "Yes" or "No", based on typical presentation.
**Reply exactly as:** `Condition:  Answer`
**Conditions:** {candidate_list}

---

Figure 15: VOI: Batch Answer Simulation (Medical Diagnosis)

**Direct Prompting and Confidence Thresholding**

**Opening**
User: Help me pick flights. My preferences are fixed; infer them and choose. Use your best judgement; don't ask for more info.
— SUPPORT HISTORY —
User: Which flight is best?
Flight 1: {option 1}
Flight 2: {option 2}
Flight 3: {option 3}
User: I prefer flight {1/2/3}
**NEW Round (no answer shown)**
User: Which flight is best?
Flight 1: {option 1}
Flight 2: {option 2}
Flight 3: {option 3}
**Required Output**
Model: The best option is Flight

Figure 16: The prompt used for Direct Prompting and Confidence Thresholding. Logit is extracted as measure of confidence.

**VOI — Prior over Feature States**

You are calibrating a probabilistic user model.

Feature: {feature}
History (support + any clarifying Q&A):
{history_ctx}

Based *only* on this history, estimate $P(\text{state})$ for the feature.
**Return STRICT JSON with keys exactly** {{states}} **that sum to 1**. Example: {"lower": 0.33, "higher": 0.33, "none": 0.34}
**JSON:**

Figure 17: Prior Estimation for VOI (Airline Preference Matching)

**VOI — Posterior with Options**

You are calibrating a probabilistic user model.

Feature: {feature}
History (support + any clarifying Q&A):
{history_ctx}

Current options:
A) {option A}
B) {option B}
C) {option C}

Estimate the *posterior* distribution over the user's {feature} state given full history.
**Return STRICT JSON with keys exactly** {{states}} **that sum to 1**.
**JSON:**

Figure 18: Posterior Estimation with Options (Airline Preference Matching)

**VOI — Candidate Preference Questions**

You are an AI assistant helping a user choose between flight options A, B, and C. You've analyzed the support examples but still have some uncertainty.

{support_history}
{qa_context}

Generate one multiple-choice question about a single aspect of the user's preference that will help decide among the options below.

A) {option A}
B) {option B}
C) {option C}

**Question:**

Figure 19: VOI Candidate Questions (Airline Preference Matching)