# OpenReview forum: "When Should AI Ask: Decision-theoretic Adaptive Communication for LLM Agents"
_ICLR.cc/2026/Conference — ICLR 2026 Conference Withdrawn Submission_

### Official Review · Reviewer_pAan · 2025-10-27

**Soundness:** 1
**Presentation:** 3
**Contribution:** 1
**Rating:** 2
**Confidence:** 5

**Summary:**

This paper introduces a Value of Information (VoI) method that enables LLM agents to adaptively decide when to ask for clarification. The proposed method forms a formal criterion based on three contextual factors: query ambiguity, task risk, and user cognitive load. The experiments show that the method outperforms standard methods across various tasks without task-specific tuning.

While this method is simple and train-free, it does suffer from several limitations (see weaknesses)

**Strengths:**

1. The method in this paper is simple and easy-to-follow
2. A train-free method to estimate when to ask clarifying questions at test time

**Weaknesses:**

1. **Limited Novelty and Limited Evaluation**. The paper's central idea of balancing the benefits and costs of clarification is not novel and has been extensively explored in prior work. As a result, the paper's central contribution feels incremental and lacks sufficient validation.
- On one hand, the main novelty of this paper rests on subdividing the "cost" into task risk and user cognitive load. Despite this, the authors offer neither theoretical arguments nor experimental evidence to demonstrate that this specific breakdown of cost provides any advantage over a single, unified cost metric.
- On the other hand, a key advertised advantage of the proposed method is its train-free nature, which contrasts with many learning-based alternatives. However, the experimental evaluation critically omits a direct comparison against such train-based baselines.

2. **Evaluation Setup Is Insufficiently Justified**. The method's performance appears highly sensitive to the chosen values for the utility function and cost. This raises a critical question about its practical applicability: how can one effectively tune these parameters for diverse domains, user queries, and levels of user impatience in real-world scenarios? It is unclear if the method is robust across a wide range of these parameter values, and the authors should clarify the conditions under which the proposed approach might fail or underperform.

3. The paper's central claim is about determining **when to ask** questions. However, Figure 2 presents a potential paradox: methods with a similar times of asking questions (#Rounds) achieve vastly different performance outcomes. So, it seems that the key difference between methods is not just when they ask, but also what they ask (i..e, **what-to-ask**). By not controlling for the question generation component, the experiment fails to isolate the "when-to-ask" variable, making it impossible to attribute the performance gains solely to the proposed 'when-to-ask' method

**Questions:**

1. The method's performance appears highly sensitive to the chosen values for the utility function and communication cost. But how can one effectively tune these parameters for diverse domains, user queries, and levels of user impatience in real-world scenarios? Is the proposed method robust across a wide range of these parameter values? What is the conditions under which the proposed method might fail or underperform?

2. Regarding Figure 2, why there is such a significant performance gap between methods, even when they engage in a similar number of clarification rounds (#Rounds)??

---

### Official Review · Reviewer_acog · 2025-10-31

**Soundness:** 3
**Presentation:** 3
**Contribution:** 2
**Rating:** 4
**Confidence:** 4

**Summary:**

This paper proposes a decision-theoretic framework for adaptive communication in LLM agents, based on the Value of Information (VoI). The core idea is to enable LLM agents to dynamically decide whether to ask clarifying questions or commit to an action, by explicitly weighing the expected utility gain of asking against a communication cost. The framework is evaluated across four diverse domains and shows consistent improvements over heuristic baselines without requiring task-specific threshold tuning.

**Strengths:**

1. The paper clearly motivates a practical and understudied problem: how LLM agents should balance autonomous action with proactive communication in the face of ambiguous user requests. The proposed VoI framework provides an interpretable rationale for when to ask.
2. The evaluation spans multiple domains (QA, recommendation, e-commerce), showing consistent improvement over several baselines (No Question, Confidence Thresholding, Adaptive Prompting).

**Weaknesses:**

1. While the proposed decision-theoretic framework is general, the claim of “no task-specific tuning” appears somewhat overstated. Although the VoI-based decision mechanism itself is task-agnostic, its implementation still relies on manually specified task parameters, namely the task risk (via fixed utility scales) and communication cost (via pre-set constants). These quantities are not inferred or learned from data, but rather predetermined in the experimental setup. As a result, the adaptivity of the method is conditioned on external priors instead of being fully autonomous. A clearer statement would be that the method requires no model-specific fine-tuning, but still depends on task-level parameter specification.
2. The communication cost is modeled by a linear function, which is difficult to reflect the real user's cognitive burden; similarly, the task risk is set by a fixed constant, which fails to reflect context sensitivity or adaptability.
3. Although the formulation of VoI is elegant, its practical application relies heavily on the LLM's ability to accurately estimate the belief distribution b(θ) and simulate user responses. The calibration analysis (Figure 3) shows that this is not always reliable, especially in noisy domains such as medical diagnosis, but the paper does not provide an in-depth analysis of how such miscalibration affects VoI decision-making.
4. The paper does not present the impact of risk weights or communication cost variations on the final utility and the number of questions asked. Such experiments would better demonstrate the robustness and adaptability of the framework.
5. Although Algorithm 1 mentions the "belief updater", the paper is vague about how it is specifically implemented. This lack of implementation details is not conducive to the reproduction and validation of the work.
6. The framework employs a loosely-coupled design for question generation, which limits its coherence and real-world applicability. The core VoI decision mechanism is critically dependent on the output of the question generation module `GenQ(H)`, yet the two are treated as independent. This creates a fundamental disconnect: the optimal timing for a question (decided by VoI) is theoretically dependent on which questions are available (generated by `GenQ`). In a real-world open-ended dialogue, these two components are tightly coupled—the decision to ask is influenced by the ability to formulate a good question. By not modeling this interaction, the framework's validity is constrained to settings where a high-quality, pre-vetted question set is available, and its ability to function in a fully autonomous capacity remains unproven.
7. Other Errors：
    1. The abbreviation should be consistent throughout the text, e.g., Value of Information (VoI) should pay attention to the case of the abbreviation.
    2. In the caption of Figure 1, the descriptions of subfigure (A) and subfigure (B) are reversed.
    3. Both Figure 1 and Figure 4 are raster images, which are relatively blurry. It is recommended to replace them with vector images.
    4. Line 398, The → the.

**Questions:**

1. In practical applications, task risk and communication cost can be difficult to quantify precisely in advance. What suggestions do the authors have for setting these parameters for unknown tasks?
2. Can the authors provide any theoretical insights or guarantees, for example, the expected utility improvement bound relative to a random questioning strategy?
3. What is the specific implementation mechanism of the belief updater? Could authors provide a concrete example to demonstrate how the posterior belief is derived from the prior belief, the question, and the answer?

---

### Official Review · Reviewer_kAd5 · 2025-11-04

**Soundness:** 2
**Presentation:** 2
**Contribution:** 4
**Rating:** 2
**Confidence:** 5

**Summary:**

The paper proposes a new VOI framework for LLMs that takes into account query ambiguity, task risk, and cognitive load when considering what to ask the user. The method is evaluated on 20 questions, 10 question medical diagnoses, flight recommendations, and a portion of webshop. The LLM is compared against baselines such as fixed round, adaptive threshold, and confidence-based prompts and demonstrates improvement over baselines in most scenarios.

**Strengths:**

The paper tackles an important question---when to ask a user clarification questions when performing agentic tasks---using a framework grounded in the social science literature (VoI). The bayesian model in 4.1 is a nice formalization of the paradigm and is appreciated. The proposed method seems to do relatively well against baselines.

**Weaknesses:**

1. The chosen evaluation paradigms aren't the most convincing. The LLM knows that 20 questions is a game, in which the guesser has clear objective functions (e.g., user enjoyment) that differ from answering correctly. Similarly, the doctor setting is also structured in a pretty unrealistic game-like way that could cause the model to behave differently. The authors mention that 20 Questions is "a classic guessing game with a long history as a paradigm for studying human and artificial decision-making under uncertainty", but do not provide citations.

2. Theoretical grounding is not as solid. At least in writing, the choice of modeling the trade-off via three factors isn't justified by any literature. Authors should try to argue that these three factors are comprehensive to a certain extent. It's also not completely clear in writing how these three factors translate to the rational model provided in section 4.1, although the rational model in itself makes sense.

3. Design decisions in experiments are arbitrary. Cost is set as a constant and arbitrarily chosen to be 0.01 or 0.05 without elaboration. Utilities are set at 1 and 10 without elaboration. The "real doctor-patient chat histories as input" mentioned in line 258 are never elaborated upon. The known candidate set is also a deviation that needs at least another 1-sentence justification. Model selection GPT-4.1 and Gemini-2.5-Flash is arbitrary and not comprehensive.

3. Presentation needs work. figure 1 is blurry and example C in it doesn't make sense -- why would a non-stop flight that is cheaper even be a question? I feel like it's more useless than the questions in example B. Figure 2 has text cut off in various places. line 44 and 52 should be \citep not \citet. line 175 has grammar issues. Some core details are omitted in the writing, such as how algorithm 1 is actually instantiated in practice; I didn't know that the posterior is computed by the LLM rather than using mathematical methods. Claim in line 309-310 "because" is too strong and unjustified. Inverted U shapes in line 315 are not that apparent in around half of the settings in the figure. Section 4 only has 1 subsection. The discussion section is in the appendix with no mention of it in the main text.

**Questions:**

See Weaknesses.

---

### Note · Authors · 2026-01-06

I have read and agree with the venue's withdrawal policy on behalf of myself and my co-authors.